# Exploring Knowledge, Attitudes, and Behaviors Toward Antibiotics Use Among Adolescents in Southern Italy

**DOI:** 10.3390/microorganisms13020290

**Published:** 2025-01-28

**Authors:** Silvia Angelillo, Giovanna Paduano, Vincenza Sansone, Anna De Filippis, Emiliana Finamore, Concetta Paola Pelullo, Gabriella Di Giuseppe

**Affiliations:** 1Department of Experimental Medicine, University of Campania “Luigi Vanvitelli”, 80138 Naples, Italy; silvia.angelillo@studenti.unicampania.it (S.A.); giovanna.paduano@unicampania.it (G.P.); vincenza.sansone@unicampania.it (V.S.); anna.defilippis@unicampania.it (A.D.F.); 2Complex Operative Unit of Virology and Microbiology, University Hospital of Campania “Luigi Vanvitelli”, 80138 Naples, Italy; emiliana.finamore@unicampania.it; 3Department of Medical, Human Movement and Well-Being Sciences, University of Naples “Parthenope”, 80133 Naples, Italy; concettapaola.pelullo@uniparthenope.it

**Keywords:** adolescent, antibiotics, antimicrobial resistance, attitudes, behaviors, Italy, knowledge, survey

## Abstract

Background: This study explored knowledge, attitudes, and behaviors regarding antibiotics among adolescents. Methods: The cross-sectional study was conducted from February to April 2024. Results: Overall, 65.2% of adolescents believed that antibiotics should only be used when prescribed, this was more likely among women and those who needed additional information regarding antibiotics. Women and adolescents without a parent with a chronic disease, who believed that it is not possible to stop antibiotics when symptoms improve and who disagreed that antibiotics are faster treatment for flu were more likely to prefer prescribed antibiotics. Antibiotic use without a medical prescription was reported by 37.4% of participants, and this was more likely among men and adolescents who had a parent with a chronic disease, who needed additional information regarding antibiotics, and who disagreed that antibiotics should only be used when prescribed. Adolescents who are afraid that antimicrobial resistance (AMR) may affect their health, who received previous information about AMR, who know that antibiotics are not useful for viral infection treatment, and who need additional information regarding antibiotics, were more likely to believe that it is important to talk about AMR at school. Conclusions: Interventions on AMR and antibiotic use among adolescents are needed.

## 1. Introduction

It is well known that antimicrobial resistance (AMR) still represents one of the most significant public health challenges across the world with increasing morbidity, mortality, and healthcare costs [1]. According to the World Health Organization, it is ranked among the top ten threats to global public health [2]. AMR concerns various aspects based on infection control, the transmission routes of pathogens, cost management, and behavioral aspects linked to the prevention and management of infections through the prescription and conscious use of antibiotics. The continuous spread of AMR is alarming, and it may be attributable to various factors, such as, for example, inappropriate prescriptions, self-medication, lack of knowledge about the use, the misuse and overuse of antibiotic agents in humans and animals, and the use of over-the-counter antibiotics [3,4,5].

Individuals’ level of knowledge and attitude towards antibiotic use greatly impacts their behaviors and insufficient and inconsistent knowledges are key factors driving their inappropriate use and this is of serious concern for the consequent spread of AMR. This information is crucial to implementing effective prevention strategies and activities, including a comprehensive multidisciplinary program involving healthcare professionals, patients, and policymakers. Several surveys on knowledge, attitudes, and behaviors have previously been conducted around the world to assess antibiotic use among specific groups, including the general population [6,7], healthcare workers [8,9,10], students [11,12], and parents [13,14,15]. However, this complex topic has not been well-studied among adolescents and, therefore, it is important to focus the attention on the young generation to promote tailor interventions to address their specific needs for the effective and efficient use of antibiotics in the future. Given this lack of information, there is a need to cover this gap and to have comprehensive knowledge and an understanding of the phenomenon in this group is imperative. In Italy, antibiotics can be sold only through a prescription, evidencing the pharmacist that the patient is being treated with the antibiotics. Despite this, people at home may take, without prescription, antibiotics from previous therapies or taken by other family members. Consequently, the aim of the present survey was firstly to investigate the level of knowledge, attitudes, and behaviors regarding antibiotics among a sample of adolescents in Italy and, secondly, to identify the various factors contributing to the main outcomes of interest.

## 2. Material and Methods

### 2.1. Setting and Participants

This observational study was part of a larger project exploring adolescents’ knowledge, attitudes, and lifestyles based on the One-Health approach [16]. The data were collected between February and April 2024 among students aged 14 to 19 years attending public middle high schools in the Campania region, Southern Italy. The sample was selected through a two-stage cluster sampling procedure. In the first stage, three high schools were randomly selected from the records that reported all regional public schools. In the second stage, adolescents were casually chosen from each school using a simple random procedure.

### 2.2. Sample Size

The minimum sample size was established before the study began, accepting a 95% confidence level, a margin of error of 5%, and a response rate of 95% [17] and assuming that 40% of adolescents took antibiotics without a medical prescription [18]. Therefore, the minimum sample size was likely to be 388 participants. To improve the precision of the results, a larger sample of participants was recruited.

### 2.3. Data Collection

Before starting the survey, a member of the research team contacted the directors of each school to plan a meeting to discuss the project purposes, the procedures for data collection and analysis, and to organize the times and modes to propose the questionnaires in the classrooms. For students < 18 years of age, enrolling in this study needed permission from their parents or legal guardians. Therefore, a member of the research team went into the classrooms and gave adolescents a sealed envelope to take home containing a cover letter, explaining the purpose and aims of the survey, that privacy and anonymity were strictly ensured, and that no payments or gifts were provided for participation. On a second planned day, the research team went to the school and, once ensured that consent had been acquired, distributed the questionnaires to students. In each classroom, there was a trained researcher who provided verbal instructions for accurate completion, helped with any doubts, and highlighted the importance of truthful responses, explaining that data would be analyzed with aggregated approaches to protect anonymity. Adolescents were free to conclude their participation at any time without explanation and, while completing the questionnaire, expressed their assent. 

### 2.4. Survey Instrument

Information about adolescents’ knowledge and behaviors related to antibiotic use and resistance was collected using a questionnaire based on previous surveys and modified to adapt the questions to our context [17,19,20,21,22]. The self-administered designed questionnaire contained closed and open-ended questions. Open-ended questions were codified, and all responses were evaluated using categorical variables. The questionnaire consists of five sections: (i) Adolescents’ socio-demographic and health-related characteristics (age, gender, nationality, parents’ education level, and parents’ occupation, chronic medical conditions, etc.); (ii) Participants’ knowledge regarding antibiotic use, with twelve true or false questions, other two questions investigating if they have heard about antibiotics and AMR with two options (“yes” or “no”) and, if yes, to specify the source of information by selecting multiple choice answers; (iii) The behavior of adolescents regarding antibiotic use with a question asking whether they have never used antibiotics with two options (“yes” or “no”) and the reasons with the possibility to select multiple answers. Adolescents were also asked if they had taken antibiotics without a medical prescription with two options (“yes” or “no”) and then to indicate the reasons for having or not taking antibiotics without a medical prescription by selecting multiple answers; (iv) Adolescents’ attitudes regarding antibiotics. One question explored their willingness to take antibiotics without the medical prescription with two options (“yes” or “no”), indicating the reasons for whether or not to take antibiotics without the medical prescription and selecting multiple answers. One question asked how much they think it is important to speak of AMR at school, on a Likert-type scale ranging from 1 (not at all important) to 10 (very much important), nine questions exploring AMR attitudes on a five-point Likert-type scale ranging from 1 (strongly disagree) to 5 (strongly agree); (v) Adolescents’ need for information about antibiotics and AMR at school.

The questionnaire was tested in a pilot study among 50 adolescents to determine its readability, comprehensibility for adolescents, and estimated completion time. Since no changes were made, the questionnaire was included in the final sample.

### 2.5. Ethics Approval

All collected data were handled according to the standards of the Declaration of Helsinki. The Ethics Committee of the Teaching Hospital of the University of Campania “Luigi Vanvitelli” authorized the study protocol (approval number 0018199/i/2024).

### 2.6. Statistical Analysis

First, descriptive statistics, analyzing means, ranges, and standard deviations for continuous variables, frequencies, and proportions for categorical variables, summarized the main characteristics of the sample. Second, the multivariate ordered logistic and logistic regression models were conducted to verify the significant relationship between dependent and independent variables. The four outcomes were: believing that antibiotics should only be used when prescribed by a physician (strongly disagree/disagree = 1; uncertain = 2; strongly agree/agree = 3) (Model 1); willingness to take an antibiotic without a medical prescription (yes = 0; no = 1) (Model 2); taking an antibiotic without a medical prescription (no = 0; yes = 1) (Model 3); believing that it is important to talk about AMR in schools (no = 0; yes = 1) (Model 4). The following independent variables have been tested for all models: age (continuous), gender (female = 0; male = 1), having both parents employed (no = 0; yes = 1); having parents with chronic medical conditions (no = 0; yes = 1); knowing that it is possible to stop taking an antibiotic as soon as symptoms improve (false = 0; true = 1); need to receive additional information about antibiotics (no = 0; yes = 1); need to receive additional information about AMR (no = 0; yes = 1); believing that antibiotics are useful to treat viral infection (false = 0; true = 1). In Model 2 and Model 3, the variable believing that antibiotics are a faster treatment for flu (strongly disagree/disagree = 1; uncertain = 2; strongly agree/agree = 3) was added. In Model 3, the variable believing that antibiotics should only be used when prescribed by a physician (strongly disagree/disagree = 1; uncertain = 2; strongly agree/agree = 3) was added. Model 3 and Model 4 included the variable concerning AMR’s capacity to affect health status (strongly disagree/disagree = 1; uncertain = 2; strongly agree/agree = 3). In Model 4, the following variables were added: the number of family members (continuous); having both parents with a university degree (no = 0; yes = 1); self-perceived health status (continuous); having a chronic medical condition (no = 0; yes = 1); having received previous information about antibiotics (no = 0; yes = 1); having received previous information about AMR (no = 0; yes = 1).

Results are presented as Odds Ratios (ORs) and 95% Confidence Intervals (CIs). All reported *p*-values are two-tailed, and a *p*-value ≤ 0.05 is evaluated as statistically significant. The significance level of the *p*-value for the inclusion and elimination of the variables in the models was set at 0.2 and 0.4, respectively. All analyses were performed with the Stata software, version 17.

## 3. Results

### 3.1. Characteristics of the Participants

Among the 531 adolescents selected, a total of 517 agreed to participate, yielding a response rate of 97.4%. The principal characteristics of the sample are presented in Table 1. A large majority of participants were female (63.4%), the average age was 17.4 years, and 21.6% reported having a chronic medical condition. Regarding their parents, 24.6% had both parents with a university degree, 71.3% had both parents employed, and 24.5% had a parent with a chronic medical condition.

### 3.2. Knowledge Related to Antibiotics and AMR

Participants were assessed on their knowledge of antibiotics and AMR based on ten statements, and the detailed responses are shown in Table 2. Of all the participants, 85.4% and 50.8% were aware of what antibiotics and AMR are, respectively. The vast majority knew that antibiotics are effective for bacterial infections (93.1%), and that can cause allergic reactions (90%). Moreover, 65.3% indicated that amoxicillin is an antibiotic, but only 54.3% indicated that paracetamol is not an antibiotic. Regarding the correct use of antibiotics, 68.2% knew that it is not possible to stop taking an antibiotic as soon as symptoms improved, and 48% mistakenly believed that it is possible to use leftover antibiotics. More than half (56.7%) mistakenly believed that antibiotics are effective for viral infections. Furthermore, 69% and 57.4% knew that antibiotics are not indicated to reduce any pain and/or inflammation and that they can kill good bacteria in the human body, respectively. Finally, only 51.5% were knowledgeable that infection prevention and control measures limit the development of AMR, and only 22.1% knew that AMR is a phenomenon whereby bacteria evolve to withstand the action of antibiotics.

### 3.3. Attitudes Toward Antibiotics, AMR, and Preventive Measures

Attitudes toward antibiotics, AMR, and preventive measures are detailed in Table 3. Almost two-thirds (65.2%) of the participants believed that antibiotics should only be used when prescribed by a physician; 31.8% and 26.5% believed that AMR is one of the top global public health threats and that researchers can help eradicate AMR before it becomes dangerous, respectively. Regarding antibiotic use, only 45.5% of adolescents agree with the statement that antibiotics are a faster treatment for flu, 27.1% wrongly believe that it is possible to stop taking the antibiotic when you start feeling better, and almost two-thirds (62.7%) of the sample agree with the statement that antibiotics should not be used in animals intended for food. Furthermore, only 32.3% of the study participants are worried that AMR may affect health.

### 3.4. Behavior Regarding Antibiotic Use

The vast majority (93.5%) had used antibiotics during their life for gastrointestinal disorders (18.9%), sore throat (18.9%), and fever (16.4%). More than one-third (37.4%) reported antibiotic use without a medical prescription and the most common reasons were non-serious problems (27.6%) and because they were on vacation (24.1%). By contrast, the main reasons among those who did not use antibiotics without a prescription (62.6%) were because they preferred to have advice from a physician (67.3%) and they did not need it (20.6%).

When asked if they would be willing to take an antibiotic without a medical prescription, 39.5% of adolescents reported that they would opt to take it. The most frequently reported reasons for this willingness were advice from a pharmacist (41.2%), and a non-serious health problem (17.5%), whereas the reasons for the 60.5% of respondents’ unwillingness to take an antibiotic without a prescription were concerns about side effects (43.2%), and trust in the physician (42.1%).

### 3.5. Sources of Information

The main source of information about antibiotics was the family (52.6%), followed by schools (18.8%), and physicians (13.8%), whereas, for AMR, the most consulted source was the school (40.2%), followed by physicians (13%).

### 3.6. Multivariate Analysis

Results of the multivariate ordered logistic and logistic regression models are shown in Table 4. When exploring attitudes, the multivariate ordered logistic regression model showed that the women and those who needed additional information regarding antibiotic use were significantly more likely to believe that antibiotics should only be used when prescribed by a physician. Those who believed that it is possible to stop taking an antibiotic as soon as symptoms improve and those who believed that antibiotics are useful to treat viral infections were significantly less likely to believe that antibiotics should only be used when prescribed by a physician.

Moreover, the results of the multivariate logistic regression model showed that women (OR = 1.57, CI 95% = 1.01–2.47), those who did not have a parent with a chronic medical condition (OR = 0.49, CI 95% = 0.31–0.81), those who believed that it is not possible to stop taking an antibiotic as soon as symptoms improve (OR = 0.51, CI 95% = 0.31–0.81), and those who disagree that antibiotics are faster treatment for flu (OR = 0.39, CI 95% = 0.21–0.71) were significantly more likely to be willing to take an antibiotic with a medical prescription.

Regarding antibiotic use without a medical prescription, the results of the multivariate logistic regression analysis indicated that four variables were significantly associated with this outcome. Men (OR = 0.56, CI 95% = 0.33–0.96), those who had a parent with a chronic medical condition (OR = 2.29, CI 95% = 1.27–4.13), those who needed additional information regarding antibiotic use (OR = 2.62, CI 95% = 1.05–6.52), those who are uncertain that AMR may affect the health status (OR = 0.74, CI 95% = 0.42–1.38), and those who disagree that antibiotics should only be used when prescribed by a physician (OR = 0.08, CI 95% = 0.04–0.14).

Moreover, those who are afraid that AMR may affect their health status (OR = 2.61, CI 95% = 1.54–4.41), those who received previous information about AMR (OR = 1.79, CI 95% = 1.04–3.09), those who know that antibiotics are not useful for viral infection treatment (OR = 0.58, CI 95% = 0.35–0.96), and those who needed additional information regarding antibiotic use (OR = 2.91, CI 95% = 1.11–7.63) were more likely to believe that it is important to talk about AMR in schools. Meanwhile, those who had a parent with a chronic medical condition (OR = 0.44, CI 95% = 0.25–0.79) were less likely to believe that it is important to talk about AMR in schools.

## 4. Discussion

To the best of our knowledge, this is the first study conducted in Italy to explore knowledge, attitudes, and behaviors toward antibiotic use among adolescents. This age represents an important period of human growth in which adolescent begin to develop their own identity; however, they are not fully capable of understanding the relationship between behavior and consequences, particularly on their health. Sometimes, from adolescence, people can also start to use antibiotics as self-medication, and these habits can persist throughout life.

Overall, our findings revealed a not-always-appropriate level of knowledge about antibiotics. Indeed, although most of the participants knew that antibiotics are medicines used to treat bacterial infections, when we indicated the name of a commonly used antibiotic, only 65.3% had correctly identified it. One notable fact is that 56.7% of respondents indicated that antibiotics are useful to treat viral infections. This value is similar to what was reported in a survey with 62% of students who believed that antibiotics were effective against viral infections [17]. Furthermore, this is similar to results reported in previous studies describing that adolescents in England did not perceive antibiotics to be specifically for bacterial infections and that they are similar to painkillers [23]. Finally, only 22.1% of respondents knew that AMR is a phenomenon whereby bacteria evolve to withstand the action of antibiotics.

Moreover, while 68.2% believe that it is possible to stop taking an antibiotic as soon as symptoms improve, a lower percentage of adolescents think that infection prevention and control measures can reduce the development of resistance (48.5%) and that it is possible to use an antibiotic to treat one illness and then another (52%). These data are worrying because they could represent behavior that amplifies the phenomenon of AMR, which could have repercussions on public health.

Furthermore, 45.5% of adolescents were of the opinion that antibiotics are a faster treatment for flu. This is confirmed by other previous and similar surveys conducted also in different settings and populations, in which people erroneously believed that antibiotics are useful for the treatment of flu [7].

Regarding attitudes towards antibiotics, 65.2% of respondents agree/strongly agree that antibiotics should only be used when prescribed by a physician. These data were comforting for the further improvement of good practice in antibiotic use.

Low risk perception was shown compared to AMR, as only 31.8% felt that AMR is one of the top global public health threats and 32.3% are afraid that AMR may affect their health. These findings are alarming and highlight the low awareness that adolescents have on the issue of AMR. A study conducted among adolescents in the UK has shown that adolescents with knowledge of AMR tended to show better attitudes toward it, and better behaviors such as taking the full course of antibiotics to not increase antibiotic resistance [24].

Regarding the use of antibiotics, 62.6% of adolescents reported antibiotic use with a medical prescription, and among these, the main reported reasons were because they preferred to have advice from a physician (67.3%) and they did not need it (20.6%). This value is lower than the 85.02% of rural and 86.02% of urban adolescents who used antibiotics prescribed by doctors [20]. These data are higher than those reported in the general population in Europe, with 22% obtaining oral antibiotic formulations without a prescription [25]. Moreover, it has been described that, in Italy, there is still a large regional variability in antibiotic consumption, which is greater in the South than in the North and the Centre; therefore, the presented results should not be representative of the national population’s behavior [26]. These data, however, confirm the important and trusted role attributed to physicians, which is essential in the relationship with patients, as reported in similar studies exploring the role of physicians in antibiotic use in other countries [27].

One surprising detail is that only 13.8% and 13% of adolescents had received information about antibiotics and AMR from physicians. This value is significantly lower compared to those reported in another survey in which 60% of adolescents had received information from physicians [17]. In this issue, a school can play an important role; indeed, it has been reported that involving students in educational programs about antibiotics and AMR with, for example, sessions of peer education, can be useful to improve knowledge and awareness of adolescents on this topic [28], as been as involving families in educational programs [29].

Intervention strategies to educate parents and healthcare workers appear important to reduce the use of antibiotics and its related risks for children and adolescents. Indeed, it has been demonstrated that early exposition to antibiotics can increase the risk of disease and allergies [30,31,32]. Further investigations on long-term trend analysis could offer valuable perspectives on how health behaviors and medication use patterns change over time, for example after a program of public health to reduce antibiotics use without prescription, as has been done for other health-related issues [33,34].

This study has several limitations that should be addressed for the interpretation of our results. First, this study was constructed on a cross-sectional design, and, therefore, it is not possible to establish the causal relationships between the determinants and the outcome of interest. Second, regarding attitudes and behaviors, adolescents may over-reported socially desirable attitudes and behaviors giving the most “desirable” answers and determining the social desirability bias, particularly given the presence of researchers and proximity to peers during the survey. This limitation was contained by assuring the confidentiality and anonymity of responses. Third, since data were collected only in the Campania Region, the sample could not be generalizable to all Italian adolescents. Finally, having completed the questionnaire at school, adolescents could have been influenced by their schoolmates. Despite the limitations described, the findings provide important information about knowledge, attitudes, and behaviors toward antibiotic use among adolescents.

## 5. Conclusions

This study contributes to highlighting a topic of great interest and relevance, emphasizing the necessity of promoting interventions on AMR and antibiotic use among adolescents. To contain this phenomenon is crucial to fill knowledge gaps and raise awareness, as well as to enhance good practice on antibiotic use among the population, particularly among this age group. Information on these topics should be added to the school career of adolescents, in a One Health approach, to prevent and respond to AMR and improve human health. Innovative approaches, including peer-to-peer education and family-centered-care meetings with schoolteachers and healthcare workers, can spread knowledge on antibiotics and AMR and improve behaviors on this public health topic.

## Figures and Tables

**Table 1 microorganisms-13-00290-t001:** Socio-demographic and anamnestic characteristics of the study population.

Characteristics	N	%
Socio-demographic		
*Age, years (continuous)*	17.4 ± 1.4 (14–21) *****	
*Gender (511)* ^a^		
Male	187	36.6
Female	324	63.4
*Nationality (513)* ^a^		
Foreigners	25	4.9
Italians	488	95.1
*Number of cohabitants (513)* ^a^	2.9 ± 0.9 (1–7)	
*Both parents with a university degree (500)* ^a^		
No	377	75.4
Yes	123	24.6
*Both parents employed (477)* ^a^		
No	137	28.7
Yes	340	71.3
Anamnestic		
*Chronic medical conditions (510)* ^a^		
No	400	78.4
Yes	110	21.6
*Parents’ chronic medical conditions (507)* ^a^		
No	383	75.5
Yes	124	24.5
*Self-perceived health status (497)* ^a^	7.6 ± 1.7 (1–10) *	

^a^ In brackets are the number of respondents to each item. * Mean ± Standard Deviation (Range).

**Table 2 microorganisms-13-00290-t002:** Knowledge about antibiotics and AMR.

Statements	Incorrect		Correct	
	N	%	N	%
Antibiotics are medicines used to treat bacterial infections (478) ^a^	33	6.9	445	93.1
Amoxicillin is an antibiotic (415) ^a^	144	34.7	271	65.3
Paracetamol is not an antibiotic (449) ^a^	205	45.7	244	54.3
Antibiotics are useful to treat bacterial infections (467) ^a^	38	8.4	429	91.9
Antibiotics are not useful to treat viral infections (466) ^a^	264	56.7	202	43.4
Antibiotics are not indicated to reduce any kind of pain and/or inflammation (464) ^a^	144	31	320	69
Antibiotics can kill the good bacteria in your bodies (448) ^a^	191	42.6	257	57.4
Antibiotics can cause allergic reactions (461) ^a^	46	10	415	90
AMR is a phenomenon whereby bacteria evolve to withstand the action of antibiotics (443) ^a^	345	77.9	345	22.1
It is not possible to stop taking an antibiotic as soon as symptoms improve (459) ^a^	146	31.8	313	68.2
Infection prevention and control measures can reduce the development of resistance (445) ^a^	229	51.5	216	48.5
It is not possible to use an antibiotic to treat one illness and then another (452) ^a^	217	48	235	52

^a^ In brackets are the number of respondents to each item.

**Table 3 microorganisms-13-00290-t003:** Attitudes towards antibiotics.

Statements	Strongly Disagree/Disagree		Uncertain		Strongly Agree/Agree	
	N	%	N	%	N	%
Antibiotics should only be used when prescribed by a physician (474) ^a^	70	14.8	95	20	309	65.2
Antibiotics should not be used in animals intended for food (466) ^a^	100	21.4	74	15.9	292	62.7
Vaccines protect against bacterial infections (467) ^a^	59	12.6	116	24.9	292	62.5
Research into new antibiotics should be encouraged (454) ^a^	30	6.6	101	22.2	323	71.2
AMR is one of the top global public health threats (434) ^a^	124	28.6	172	39.6	138	31.8
Researchers can help in eradicating antibiotic resistance before it becomes dangerous (441) ^a^	101	22.9	223	50.6	117	26.5
You are afraid that AMR may affect your health (443) ^a^	124	28	176	39.7	143	32.3
Antibiotics are a faster treatment for flu (453) ^a^	99	21.8	148	32.7	206	45.5
You can stop taking the antibiotic when you start feeling better (457) ^a^	223	48.8	110	24.1	124	27.1

^a^ Number of respondents to the statements is in brackets.

**Table 4 microorganisms-13-00290-t004:** Multivariate ordered logistic and logistic regression models to identify those predicting the above-mentioned outcomes of interest.

Model 1. Believing that antibiotics should only be used when prescribed by a physician			
Log likelihood = −320.06; χ^2^ = 32.04 (7 df), *p* < 0.001			
Variable	Coeff	95% CI	*p*
*Antibiotics are useful to treat viral infections*			
False	1 ^a^		
True	−0.71	−1.15–−0.26	0.002
*Gender*			
Male			
Female	0.64	0.21–1.07	0.004
*It is possible to stop taking antibiotics as soon as symptoms improve*			
False	1 ^a^		
True	−0.61	−1.07–−0.15	0.009
*Need to receive additional information about antibiotics*			
No	1 ^a^		
Yes	0.58	0.05–1.11	0.032
*Parents’ chronic medical conditions*			
No	1 ^a^		
Yes	−0.29	−0.78–0.18	0.221
*Both parents employed*			
No	1 ^a^		
Yes	0.28	−0.19–0.75	0.240
*Age, years (continuous)*	−0.09	−0.26–0.07	0.258
Model 2. Willingness to take an antibiotic with a medical prescription			
Log likelihood = −232.55, χ^2^ = 33.96 (7 df), *p* > 0.0001			
Variable	OR	95% CI	*p*
*Antibiotics are a faster treatment for flu*			
Strongly agree/Agree	1 ^a^		
Uncertain	0.69	0.37–1.31	0.264
Strongly disagree/disagree	0.39	0.21–0.71	0.002
*Parents’ chronic medical condition*			
No	1 ^a^		
Yes	0.49	0.31–0.81	0.004
*It is possible to stop taking antibiotics as soon as symptoms improve*			
No	1 ^a^		
Yes	0.51	0.31–0.81	0.005
*Gender*			
Male	1 ^a^		
Female	1.57	1.01–2.47	0.049
*Age, years (continuous)*	0.91	0.77–1.08	0.290
*Need to receive additional information about AMR*			
No	1 ^a^		
Yes	0.74	0.42–1.31	0.292
Model 3. Taking an antibiotic without a medical prescription			
Log likelihood = −176.01, χ^2^ = 131.89 (9 df), *p* < 0.001			
Variable	OR	95% CI	*p*
*Age, years (continuous)*	1.11	0.91–1.37	0.326
*Parents’ chronic medical condition*			
No	1 ^a^		
Yes	2.29	1.27–4.13	0.006
*Gender*			
Male	1 ^a^		
Female	0.56	0.33–0.96	0.035
*Need to receive additional information about antibiotics*			
No	1 ^a^		
Yes	2.62	1.05–6.52	0.038
*Concern that AMR may affect the health status*			
Strongly agree/Agree	1 ^a^		
Uncertain	0.74	0.42–1.28	<0.001
Strongly disagree/disagree	Backward elimination		
*Antibiotics should only be used when prescribed by a physician*			
Strongly agree/Agree	1 ^a^		
Uncertain	Backward elimination		
Strongly disagree/disagree	0.08	0.04–0.14	<0.001
*Antibiotics are a faster treatment for flu*			
Strongly agree/Agree	1 ^a^		
Uncertain	Backward elimination		
Strongly disagree/disagree	1.65	0.97–2.79	0.065
*It is possible to stop taking antibiotics as soon as symptoms improve*			
No	1 ^a^		
Yes	1.64	0.92–2.94	0.093
*Need to receive additional information about AMR*			
No	1 ^a^		
Yes	0.55	0.22–1.35	0.190
Model 4. Believing important talking of AMR in the schools			
Log likelihood = −187.96, χ^2^ = 62.52 (10 df), *p* < 0.001			
Variable	OR	95% CI	*p*
*Concern that AMR may affect the health status*			
Strongly agree/Agree	1 ^a^		
Uncertain	Backward elimination		
Strongly disagree/disagree	2.61	1.54–4.41	<0.001
*Parents’ chronic medical condition*			
No	1 ^a^		
Yes	0.44	0.25–0.79	0.006
*Need to receive additional information about antibiotics*			
No	1 ^a^		
Yes	2.91	1.11–7.63	0.030
*Antibiotics are useful to treat viral infections*			
False	1 ^a^		
True	0.58	0.35–0.96	0.034
*Having received previous information about AMR*			
False	1 ^a^		
True	1.79	1.04–3.09	0.036
*Need to receive additional information about AMR*			
No	1 ^a^		
Yes	2.39	0.86–6.63	0.094
*Gender*			
*Age, years (continuous)*	1.19	0.96–1.49	0.107
Male	1 ^a^		
Female	1.51	0.89–2.52	0.122
*Self-perceived health status (continuous)*	0.88	0.75–1.04	0.130
*Number of family members (continuous)*	0.85	0.61–1.15	0.285

^a^ Reference category.

## Data Availability

The data presented in this study are available upon request from the corresponding author due to ethical restrictions.

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
