# Peer review of "Exploring Knowledge, Attitudes, and Behaviors Toward Antibiotics Use Among Adolescents in Southern Italy"

_microorganisms, 2025, doi:10.3390/microorganisms13020290_

Round 1

Reviewer 1 Report

Comments and Suggestions for Authors

The manuscript entitled „Exploring knowledge, attitudes, and behaviors toward antibiotic use among adolescents in Southern Italy.” describes a study conducted among students aged between 14-19 years. Topic of this manuscript is a very interesting topic, and obtained results are novel and provide very useful information about knowledge, behaviour about antibiotics.

I have only some minor comments.

Comments

1) What was the source of questionaries? Were these questions collected from the study coordinators? Or were these questions taken from earlier international study? Please, clarify!

2) In the abstract „AMR” is mentioned, however, full name is missing in the abstract.

3) In Table 1., this part is unclear:

„Both parents employed (366)” is written however, „137+340” are given, but these do not make 366. Please, clarify!

Author Response

The manuscript entitled „Exploring knowledge, attitudes, and behaviors toward antibiotic use among adolescents in Southern Italy.” describes a study conducted among students aged between 14-19 years. Topic of this manuscript is a very interesting topic, and obtained results are novel and provide very useful information about knowledge, behaviour about antibiotics.

I have only some minor comments.

Comments

1) What was the source of questionaries? Were these questions collected from the study coordinators? Or were these questions taken from earlier international study? Please, clarify!

 R: The sources of the questionnaire have been added in the survey instrument paragraph. Some questions were taken from earlier international studies cited in the text. Other questions were added or modified to adapt the survey to our context.

2) In the abstract „AMR” is mentioned, however, full name is missing in the abstract.

 R: As suggested, the full name of “AMR” has been added in the abstract.

3) In Table 1., this part is unclear:

„Both parents employed (366)” is written however, „137+340” are given, but these do not make 366. Please, clarify!

R: We apologize for the mistake, the parents’ employed number has been corrected in Table 1.

Reviewer 2 Report

Comments and Suggestions for Authors

I have read this paper with interest, and there is some qualitative type of value in the current report. I have read this as an non-Italian, with a background on pediatric pharmacology, including but not limited to antibiotics.

I only realized after re-reading that antibiotics seem to be available without prescription ? we for sure need some more information on this aspect (like how, type of antibiotics, etc).

Specific suggestions

I would suggest to reconsider the title, as the questionnaire seems to be beyond antibiotic use (cfr first line of the abstract), so that I would suggest to align the title to the abstract (antibiotics, instead of antibiotic use).

Line 17… not possible to stop antibiotic…

Line 19, please check language, I assume you mean that were more likely to prefer prescribed antibiotics ?

Line 26, more likely to believe that it is important to talk about AMR at school ?

Line 34. Suggest to start an new sentence by According to the …

In the current version, there seems to be insufficient attention for the assent of the adolescents. This is likely more relevant as the recruitment have been done in schools, so that potential limitations to abstain may occur. Could you please comment on this aspect of the trial ?

How has the survey instrument been constructed and validated ? I miss methodological information like content validation or similar, and why has a 3 level Likert scale been used instead of eg a 4 or 5 subclass approach, perceived to be more common by this reviewer.

Why was the sample shifted to female participants ?

Discussion

I would suggest to explore the available literature, like eg PMID 28592579, PMID 35399743, PMID 32325791 to add perspective to the work reported (I have no conflict of interest, and I’m neither involved in any of these papers).

Author Response

I have read this paper with interest, and there is some qualitative type of value in the current report. I have read this as an non-Italian, with a background on pediatric pharmacology, including but not limited to antibiotics.

1) I only realized after re-reading that antibiotics seem to be available without prescription? we for sure need some more information on this aspect (like how, type of antibiotics, etc).

R: The Decree 31 March 2008 of the Ministry of Health "Delivery by the pharmacist, in case of urgency, medicines with prescription without presentation of the prescription" provides that antibiotics can be sold only on the prescription, evidencing the pharmacist that the patient is being treated with the antibiotics, Such as: a) the presence in the pharmacy of a medical prescription issued on a date which suggests that the patient is still being treated with the medicinal product requested; b) the presentation by the customer of an unusable packaging, for example, a damaged vial.

Despite this, people at home may take, without prescription, antibiotics from previous therapies or taken by other family members (line 57-59).

2) Specific suggestions

I would suggest to reconsider the title, as the questionnaire seems to be beyond antibiotic use (cfr first line of the abstract), so that I would suggest to align the title to the abstract (antibiotics, instead of antibiotic use).

R: As suggested, we modified the manuscript according to the reviewer.

Line 17… not possible to stop antibiotic…

R: As suggested, we modified the manuscript according to the reviewer.

Line 19, please check language, I assume you mean that were more likely to prefer prescribed antibiotics ?

R: As suggested, we modified the manuscript according to the reviewer.

Line 26, more likely to believe that it is important to talk about AMR at school ?

R: As suggested, we modified the manuscript according to the reviewer.

Line 34. Suggest to start an new sentence by According to the …

R: As suggested, we modified the manuscript according to the reviewer.

3) In the current version, there seems to be insufficient attention for the assent of the adolescents. This is likely more relevant as the recruitment have been done in schools, so that potential limitations to abstain may occur. Could you please comment on this aspect of the trial?

R: As suggested, we added a sentence explaining that adolescents expressed their assent to completing the questionnaire at the end of the data collection paragraph. (pag 3, line 94-95)

4) How has the survey instrument been constructed and validated? I miss methodological information like content validation or similar, and why has a 3 level Likert scale been used instead of eg a 4 or 5 subclass approach, perceived to be more common by this reviewer.

R: After conducting the pilot study, the reliability of the questionnaire was evaluated through Cronbach’s alpha, which indicated a good internal consistency with a value of 0.75.

Moreover, AMR attitudes were explored with a five-points Likert-type scale ranging from 1 to 5 as reported in the survey instrument paragraph (1 was strongly disagree, 2 was disagree, 3 for uncertain, 4 was agree and 5 was strongly agree). Performing statistical analysis, “strongly disagree” and “disagree” has been linked, such as “agree” and “strongly agree”.

5) Why was the sample shifted to female participant?

R: As reported in the setting and participants paragraph, the sample was selected through a two-stage cluster sampling procedure, and adolescents were casually chosen using a simple random procedure. Therefore, the sample shift to a major female participation appears casual.

Discussion

6) I would suggest to explore the available literature, like eg PMID 28592579, PMID 35399743, PMID 32325791 to add perspective to the work reported (I have no conflict of interest, and I’m neither involved in any of these papers).

R: As suggested, a better explanation of the available literature has been performed and other perspectives have been added in the Discussion section.

Reviewer 3 Report

Comments and Suggestions for Authors

Dear authors,

I have now completed the review of the manuscript titled "Exploring knowledge, attitudes, and behaviors toward antibiotic use among adolescents in Southern Italy."

In the present study, the authors employed a robust two-stage cluster sampling procedure to select participants, which helps enhance representativeness. The researchers also calculated an appropriate minimum sample size using established statistical parameters and achieved a strong response rate of 97.4%.

The manuscript is interesting and, in general, fairly well-written.

However, I still have some suggestions to further improve the quality of the manuscript.

1. The geographical restriction to the Campania region raises concerns about external validity. While the authors note this limitation, they do not adequately discuss how regional factors might influence their results or affect generalizability to other Italian populations. I would like to recommend authors to discuss other articles, which would provide valuable comparative data about early-life antibiotic exposure and its long-term health impacts. For example, since authors' paper discusses adolescent antibiotic use patterns, breifly discussing article like Prenatal and postnatal exposure to antibiotics and risk of food allergy in the offspring: a nationwide birth cohort study in South Korea understanding how early exposure affects later health outcomes could provide important context for intervention strategies.

2. Social desirability bias presents another major concern. Although the researchers attempted to mitigate this by ensuring confidentiality, the school-based data collection setting likely influenced responses. Students may have felt pressure to provide "correct" answers, particularly given the presence of researchers and proximity to peers.

3. The statistical analysis appears sound, using appropriate multivariate regression models. However, the presentation of results could be improved. The authors report numerous statistical associations without sufficient discussion of their practical significance or potential mechanisms. Some concerning findings warrant deeper exploration. For instance, the fact that 37.4% of adolescents reported using antibiotics without prescription represents a serious public health issue that deserved more thorough analysis of contributing factors and potential interventions.

4. While the authors position this as the first such study in Italy among adolescents, they could have better contextualized their findings within the broader literature on antibiotic use among young people globally. The discussion section misses opportunities to draw meaningful comparisons with similar studies in other countries. For example, discussing other countries' long-term trend analysis, like National Trends in Allergic Rhinitis and Chronic Rhinosinusitis and COVID-19 Pandemic-Related Factors in South Korea, from 1998 to 2021, could offer valuable perspectives on how health behaviors and medication use patterns change over time, which relates to findings about antibiotic use behaviors among adolescents.

5. The recommendations section is underdeveloped. While the authors suggest incorporating antibiotic resistance education into school curricula, they provide little concrete guidance on implementation or evidence-based strategies for improving adolescent antibiotic practices.

Thank you for your valuable contributions to our field of research. I look forward to receiving the revised manuscript.

Author Response

Dear authors,

I have now completed the review of the manuscript titled "Exploring knowledge, attitudes, and behaviors toward antibiotic use among adolescents in Southern Italy."

 In the present study, the authors employed a robust two-stage cluster sampling procedure to select participants, which helps enhance representativeness. The researchers also calculated an appropriate minimum sample size using established statistical parameters and achieved a strong response rate of 97.4%.

 The manuscript is interesting and, in general, fairly well-written.

 However, I still have some suggestions to further improve the quality of the manuscript.

  1. The geographical restriction to the Campania region raises concerns about external validity. While the authors note this limitation, they do not adequately discuss how regional factors might influence their results or affect generalizability to other Italian populations. I would like to recommend authors to discuss other articles, which would provide valuable comparative data about early-life antibiotic exposure and its long-term health impacts. For example, since authors' paper discusses adolescent antibiotic use patterns, breifly discussing article like Prenatal and postnatal exposure to antibiotics and risk of food allergy in the offspring: a nationwide birth cohort study in South Korea understanding how early exposure affects later health outcomes could provide important context for intervention strategies.

R: The geographical restriction to a region of Southern Italy has been discussed in relation to national data on antibiotic consumption. Furthermore, the characteristics of the population surveyed is similar to that of the other Italian regions, as already evidenced in other works conducted in the same geographical area and with the same methodology. Moreover, a sentence on early-life antibiotic exposure and intervention strategies have been added in the Discussion section.

  1. Social desirability bias presents another major concern. Although the researchers attempted to mitigate this by ensuring confidentiality, the school-based data collection setting likely influenced responses. Students may have felt pressure to provide "correct" answers, particularly given the presence of researchers and proximity to peers.

R: This issue had already been described in the Limitation section. Moreover, ensuring anonymity and clarifying that the questionnaire was not subject to a personal assessment, was a way of containing this limit, as it has already been used in previous published studies, despite the school background and presence of researchers. Nevertheless, as suggested, this point has been better explained.

  1. The statistical analysis appears sound, using appropriate multivariate regression models. However, the presentation of results could be improved. The authors report numerous statistical associations without sufficient discussion of their practical significance or potential mechanisms. Some concerning findings warrant deeper exploration. For instance, the fact that 37.4% of adolescents reported using antibiotics without prescription represents a serious public health issue that deserved more thorough analysis of contributing factors and potential interventions.

R: As suggested, some findings have been better explained in the Discussion section and potential interventions were proposed.

  1. While the authors position this as the first such study in Italy among adolescents, they could have better contextualized their findings within the broader literature on antibiotic use among young people globally. The discussion section misses opportunities to draw meaningful comparisons with similar studies in other countries. For example, discussing other countries' long-term trend analysis, like National Trends in Allergic Rhinitis and Chronic Rhinosinusitis and COVID-19 Pandemic-Related Factors in South Korea, from 1998 to 2021, could offer valuable perspectives on how health behaviors and medication use patterns change over time, which relates to findings about antibiotic use behaviors among adolescents.

R: As suggested, comparison with similar studies in other countries are been added. Moreover, a sentence on the perspective of a long-term study on antibiotic use has been included in the Discussion section.

  1. The recommendations section is underdeveloped. While the authors suggest incorporating antibiotic resistance education into school curricula, they provide little concrete guidance on implementation or evidence-based strategies for improving adolescent antibiotic practices.

 R: A better explanation on how implement strategies about antibiotic practices has been added in the Discussion section.

Round 2

Reviewer 3 Report

Comments and Suggestions for Authors

All comments were addressed.